# Spatial Panel Data Analysis on the Relationship between Provincial Economic Status and Enrolment in the Social Security Scheme amongst Migrant Workers in Thailand, 2015–2018

**DOI:** 10.3390/ijerph19010181

**Published:** 2021-12-24

**Authors:** Shaheda Viriyathorn, Mathudara Phaiyarom, Putthipanya Rueangsom, Rapeepong Suphanchaimat

**Affiliations:** 1International Health Policy Program, Ministry of Public Health, Tiwanon Rd., Nonthaburi 11000, Thailand; mathudara@ihpp.thaigov.net (M.P.); putthipanya@ihpp.thaigov.net (P.R.); rapeepong@ihpp.thaigov.net (R.S.); 2Department of Disease Control, Division of Epidemiology, Ministry of Public Health, Tiwanon Rd., Nonthaburi 11000, Thailand

**Keywords:** insured migrants, negative binomial regression, spatiotemporal regression, gross provincial product

## Abstract

Background: Thailand has a large flow of migrants from neighbouring countries; however, the relationship between economic status at the provincial level and the insured status of migrants is still vague. This study aimed to examine the association between provincial economy and the coverage of the Social Security Scheme (SSS) for migrants. Methods: Time-series data were analysed. The units of analysis were 77 provinces during 2015–2018. Data were obtained from the Social Security Office (SSO). Spatiotemporal regression (Spatial Durbin model (SDM)) was applied. Results: Migrant workers were mostly concentrated in Greater Bangkok, the capital city and areas surrounding it, but SSS coverage was less than 50%. However, the ratio of insured migrants to all migrants seemed to have positive relationship with the provincial economy in SDM. The ratio of insured migrants to all migrants was enlarged in all regions outside Greater Bangkok with statistical significance. Conclusions: Low enforcement on employment law in some areas, particularly Greater Bangkok, can result in lesser SSS coverage. The provincial economic prosperity did not guarantee large SSS coverage. Interventions to ensure strict insurance enrolment are required.

## 1. Introduction

Migration is a global phenomenon due to several factors such as economic opportunities, convenience of transportation, political and religious conflicts, and even human trafficking [1,2]. In 2019 there were an estimated 272 million international migrants worldwide [3]. This number is predicted to exceed 405 million in the next three decades [2]. Migrant health is, therefore, recognized as a global public health priority due to a concern of the spread of infectious diseases [4] and concern over the lack of resources in the destination countries to address the high influx of migrants. Problems have been aggravated given the low rates of health insurance coverage and high barriers to access health services amongst migrants in the receiving countries.

Southeast Asia is one of the most dynamic regions in the world [5]. Among many countries in the region, Thailand plays an important role in international migration and is the destination of a large flow of migrants from Cambodia, Lao PDR, Myanmar, and Vietnam, called CLMV nations [6]. Thailand’s Migration Report 2019 estimated that 4.9 million non-Thais reside in the country [7]. As of October 2021, registered migrants accounted for 2.3 million people (3.5% of the Thai population) [8,9]. Most of them were low-skilled migrant workers from Cambodia (19.4% of total registered migrants), Lao PDR (9.1%), Myanmar (62.3%), and Vietnam (0.006%) [8].

The International Labour Organization recently suggested that migrants contributed to about 4.3–6.6% of Thai gross domestic product (GDP) [10]. The Thai government implemented measures to protect the health of migrant workers. Registered migrants working in the formal private sector are entitled to the Social Security Scheme managed by the Ministry of Labour [11]. This scheme is financed by payroll tax through the tripartite contribution, equally from employers, employees, and the government [12]. Both Thais and migrant workers have equal rights of access to the benefits of the SSS [12], which covers inpatient (IP) care, outpatient (OP) care, high-cost treatment, and additional fringe benefits, such as pension and unemployment allowance [11]. SSS members can access health services in specific registered providers, which cover all public hospitals and a few contracted private hospitals. In 2019, the number of migrant workers insured with the SSS was approximately 1.2 million [13]. Undocumented migrants and those who work in the informal sector can access public migrant health insurance managed by the Ministry of Public Health, namely, the Health Insurance Card Scheme (HICS). HICS is a semi-voluntary public prepayment scheme financed by the purchase of the annual premium by applicants [11,12]. It is not recognized as a fully mandatory insurance scheme as those eligible to be insured need to register with the government first (turning from undocumented status to documented status), but there is no penalty for those that fail to be insured or those denying insurance enrolment.

Despite existing evidence showing that the Thai economy has been shaped by the migrant labour force [11,14], very little is known about the relationship between economic status at the provincial level and the insured status of migrants. Therefore, the objective of this study was to examine the association between the provincial economy and the insured status of migrants in a province, while taking into account the influence of time trends and geographical differences.

## 2. Materials and Methods

### 2.1. Study Design

We used quantitative secondary data analysis on time-series data during 2015–2018. The units of analysis were provinces. There are 77 provinces in Thailand. This meant that we analysed a total of 308 records (77 provinces × 4 years). Data in 2019 and onward were not sufficiently complete to perform analysis.

### 2.2. Data Sources and Variables

The interested variables are as follows: (1) the number of insured migrants (SSS migrants) from the Social Security Office; and (2) the number of migrants acquiring work permits from the Department of Employment (DOE) of the Ministry of Labour. As migrants holding work permits can be the workers in either the informal or formal sectors, we focused on the two indicators: (1) the ratio of insured migrants per the volume of formal-sector migrants; and (2) the ratio of insured migrants per volume of total migrants. We preferred the term ratio over the term proportion as the numerator (SSS migrants) and the denominator (formal-sector migrants or total migrants having work permits) were retrieved from different sources. These two indicators served as the main dependent variables of this study.

The main independent variables encompassed gross provincial product (GPP) per capita (in Thai Baht). Other covariates included a dummy variable that represented the country’s regions (Greater Bangkok, Central, North-eastern, Northern, Southern, Eastern, and Western); the percentage contribution of the selected businesses to GPP per capita including the percentage contribution of agriculture business to GPP per capita (continuous variable); the percentage contribution of manufacturing business to GPP per capita (continuous variable); the percentage contribution of construction to GPP per capita (continuous variable); the percentage contribution of wholesale and retail trade to GPP per capita (continuous variable); and the percentage contribution of accommodation and food service to GPP per capita (continuous variable). The contribution of these businesses to GPP per capita was estimated by dividing monetary values of these businesses by the overall GPP per capita in a particular province. These businesses were purposively selected by considering businesses that mostly required migrant workers to operate. These data were acquired from the Office of the National Economic and Social Development Council (NESDC). GPP per capita was selected based on the assumption that it could be a proxy for provincial economic level, while the economic contributions of various business sectors were selected, as they possibly influenced the employment rate of migrant workers.

### 2.3. Data Analysis

We performed both descriptive statistics and econometric analysis. In the first part, the annual growth rate of the number of migrants was presented. As data from the SSO and the DOE were stored as monthly records but economic data were stored as yearly records, we transformed migrant volume data from monthly records to yearly records. We used the median of migrants’ data monthly records as a proxy for migrants’ data in a respective year.

In the second part, we started with data visualisation by plotting the ratio of insured migrants against GPP per capita (in log scale). The regression slope of the plots was calculated. We then applied econometric techniques, namely negative binomial (NB) regression and spatiotemporal regression (spatial Durbin model (SDM)) to account for the influence of space and time. Spatiotemporal regression was applied as we hypothesised that in reality there was always mobility of migrants across provinces, which were adjacent to each other. Moreover, there was a possibility that one province might have a spillover effect to nearby provinces as it might closely interact with the surrounding provinces through population flow, information flow, and logistic flow.

SDM was set as Y_t_ = ρWY_t_ + αι_N_ + X_t_β + WX_t_θ + u_t_, where u_t_ = λW u_t_ + ε_t_ [15]. The dependent variable Y_t_ denoted an Nx1 vector comprising one observation for every unit in the sample (i = 1, …, N) at time t (t = 1, …, T). An N × 1 vector of ones associated with α, the constant term parameter, was represented by ι_N_. The variable X_t_ denotes N × K metrics of exogenous independent variables at time t which were associated with parameter β presented in the form of K × 1 vector. The N × 1 vector, u_t_, referred to spatial random effects, and λ indicated the spatial autocorrelation coefficient. W was an N × N metric with non-negative values that showed the pattern of an interaction effect between sample units. The error term, ε_t_, was contained in the N × 1 vector [15,16]. Stata version 14 (serial number: 401406358220) was used to analyse data.

Queen contiguity matrix was employed for SDM, and robust standard error was used. The final results were presented in the forms of incidence rate ratio (IRR) and its 95% confidence interval (CI).

### 2.4. Model Fitness Assessment

In order to evaluate the goodness of fit of the models, four key measures were analysed: (1) mean absolute error (MAE); (2) mean squared error (MSE); (3) root mean squared error (RMSE); and (4) mean absolute percent error (MAPE). Akaike Information Criterion (AIC) and the Kernel density graphs were used to assess the goodness of fit of all models.

## 3. Results

### 3.1. Descriptive Analysis

During 2015–2018, the number of all migrants in all regions had changed over time. Greater Bangkok had the largest number of migrant workers, growing from 656,725 in 2015 to 1,248,091 workers in 2018. Despite a small number of migrants compared with Greater BKK, the northeastern region had the highest annual growth rate (the slope of the number of migrants plotted against the study years), which accounted for 40%, while the negative growth rate was found in the northern region (−1%). Similar patterns were expressed in formal-sector migrants. The highest growth rate at 77% was observed in the western region. The number of migrants in the formal sector insured with SSS rose remarkably in all regions, including Greater Bangkok. The southern and western regions showed over 60% increment of the average annual growth rate. Other regions saw the growth rate of SSS migrants by approximately 40% (Table 1).

The ratio of SSS insurees to formal-sector migrants was altered during the study years. The central and the northern regions seemed to have the greatest density of insured migrants. However, the ratio in central and eastern regions experienced a decreasing trend as time passed by while the western and southern regions showed an increasing trend. Regarding the ratio of SSS insurees to all migrants, there was no specific clustering pattern between 2015 and 2017. Nonetheless, in 2018, the choropleth map displayed a darkening area (referring to the enhancement in the ratio) in the central and western regions and some provinces in the southern region (Figure 1 and Figure 2).

By exploring the provincial level, the median values of the ratio of insured migrants to formal-sector migrants varied between 0.8 and 1.2, while the median ratio of insured migrants to all migrants remained stable at 0.3–0.4. GPP per capita continuously increased from THB 146,575.7 (USD 4536.5) in 2015 to THB 169,506.6 (USD 5246.2) in 2018. The percentage fraction of agriculture, forestry, and fishing to GPP per capita gradually declined from 15.6% to 14.8% during the period observed. The manufacturing business contributed the most to GPP per capita, expanding from 18.8% in 2015 to 19.1% in 2018 with some fluctuations, whereas the share of construction slightly fluctuated around 3.1–3.2%. Overtime, the economic fraction of wholesale and retail trade and repair of motor vehicles enlarged together with accommodation and food service activities. The median economic contribution of wholesale business grew from 11.5% to 13.0%, while that of accommodation and food service increased from 2.7% to 3.4% in the study years, Table 2.

Figure 3 and Figure 4 plotted the ratio of insured migrants to formal-sector migrants and to the total number of migrants against GPP per capita in the natural log scale. A relatively flat regression slope in Figure 3 inferred that provincial economic status did not show a significant relationship with the ratio of insured migrants to formal-sector migrants. A positive regression slope in Figure 4 implied that the ratio of insured migrants to all migrants appeared to increase mostly in the economically better-off provinces; however, statistical significance was not found as the corresponding 95% CI covered zero value.

### 3.2. Negative Binomial (NB) Regression

Table 3 shows the results of NB regression. Provincial prosperity did not demonstrate a significant relationship with the ratio of insured migrants to formal-sector migrants. A positive relationship was found in certain variables. The coverage of SSS amongst formal-sector migrants in the southern region was almost five times larger than the coverage in Greater Bangkok. A significant association was also found in the western region (IRR = 3.0 (95% CI = 1.6, 5.9) and northern region (IRR = 2.2 (95% CI = 1.2, 4.3)). The proportion of SSS migrants in the formal sector dwindled by 1.7% for a 1% enlargement of the agriculture, forestry, and fishing industries to GPP per capita. There was a 5.4% decrease for a 1% increment of the economic contribution by accommodation and food services. A 1% increment of the contribution of wholesale and retail trade and repair of motor vehicles to GPP per capita resulted in a 2.5% increase in insured migrants in the formal sector.

Concerning the ratio of insured migrants to all migrants, the provincial economy did not display a significant association with SSS coverage. The central region and western region displayed a larger proportion of insured migrants than Greater Bangkok (IRR = 2.7 (95% CI = 1.6, 4.3) for the central region and IRR = 2.2 (95% CI =1.3, 3.7) for the western region). The northern and eastern regions demonstrated a significantly positive association with SSS coverage (by about 87.2% and 65.1% increase in SSS coverage compared with Greater Bangkok, respectively). Note that no statistical difference was found in the contribution of all types of industries to provincial GPP per capita.

### 3.3. Spatiotemporal Regression

In the angle of the ratio of the SSS migrants to formal-sector migrants, GPP per capita displayed a less-than-one IRR which reflected the negative relationship with the expected indicator notwithstanding that no statistical significance was observed. However, this relationship turned out to be significantly positive in the ratio of insured migrants to all migrants. A one-point increment of the GPP per capita in log-Baht unit was related to the enhancement of SSS coverage amongst all migrants by approximately 68.2% (IRR = 1.682 (95% CI = 1.1, 2.7)). The ratio of insured migrants to all migrants appeared to be enhanced outside Greater Bangkok. By using Greater Bangkok as a reference, the IRR of other regions varied between 3.9 and 11.3 with statistical significances for all. Agricultural business, construction, and accommodation services showed a negative relationship (less-than-one IRR) with SSS coverage amongst all migrants, while a positive relationship was observed in manufacturing and wholesale businesses. Nevertheless, all of these economic variables did not exhibit statistical significances, see Table 4.

### 3.4. Goodness-of-Fit Check

By a visualisation on Kernel density plot, spatiotemporal regression seemed to be fitter with actual data than NB regression for both indicators (Figure 5 and Figure 6). The values of MAE, MSE, RMSE, and MAPE confirmed this observation as most of the values were smaller in spatiotemporal regression compared with NB regression. AIC in the spatiotemporal regression was approximately eight to nine-times lower than in NB regression; implying a better goodness of fit in SDM (Table 5).

## 4. Discussion

Overall, this research study is one of the first studies that seeks to explore the relationship between provincial economy and enrolment of migrants to insurance. The positive relationship between GPP per capita and SSS enrolment amongst migrant workers in all sectors was observed in spatiotemporal regression, the fittest model amongst all interested models (every log-Baht of GPP per capita expanded the SSS coverage by approximately 68%). However, SSS coverage amongst formal-sector migrants did not show statistical significance.

The remarkable divergences of health insurance coverage were observed across geographical regions. Formal sector migrants in the southern and western regions seemed to have more opportunity to be insured than the other regions, when compared with Greater Bangkok. This discovery alludes to the fact that, although Greater Bangkok is the economic centre of the country where law enforcement on the employment law is supposed to be stringent, the ratio of SSS insurees to either migrants in the formal sector or all migrants was relatively low in comparison to other regions. This also pointed to a relatively weak enforcement of employment law in Bangkok or the proportion of migrants in the informal sector in Bangkok was relatively huge, resulting in low SSS coverage of all migrants compared with other parts of the country.

The Social Security Act 2015 [17] indicates that employers must have their wage deducted in the same proportion of migrants as part of payroll contribution. However, a qualitative study by Kunpeuk et al., pointed out that some employers elude the law by not contributing to payroll tax, and this meant that their migrant workers were uninsured [18]. Aside from this breach of law by employers, paying for health insurance premiums was avoided by some migrant employees, especially those in good health who found that insurance provided little benefit compared with their health needs [18]. This situation was also found in China where uninsured migrants expressed that they did not want to join a health insurance plan because paying for health services by out-of-pocket was a better approach economically than making regular payments [19].

Every percentage of agricultural business, construction, and accommodation services relative to GPP per capita resulted in a decrease in SSS coverage. Despite showing no statistical significance, the positive relationship between the contribution of some businesses and insurance coverage aligned with basic knowledge that areas with more formal industries tended to hire migrants at a large scale, which thus resulted in greater SSS coverage. In contrast, a negative relationship was found in agricultural, construction, and accommodation businesses, which mostly relied on informal workers. According to the Thailand Migrant Report in 2014, the percentage of registered migrants was higher in well-established factories than in non-factory settings, such as agricultural, fisheries, and construction industries [20]. Moreover, some migrants in the agricultural sector received wages on seasonal or daily basis [7]. This population group is not mandatorily covered by the SSS.

As briefly mentioned above, it is quite surprising that SSS coverage for migrants in Greater Bangkok was less powerful compared with other regions. A larger share of migrants insured with SSS seemed to occur outside Greater Bangkok although more than one-third (36%) of work permits were issued in Greater Bangkok [7]. Apart from the reason that a greater proportion of migrants in Grater Bangkok are engaged in the informal sector, another possibility is that the features of work in industries in Greater Bangkok are complex and do not have fixed boundaries. This is because Greater Bangkok is the main economic city having a large number of migrant workers [21] that commute from and to nearby provinces as day workers. This might explain the difficulties of regulating insurance coverage for work permit holders. Moreover, the number of migrants in some regions such as the northeastern region was relatively low compared with Greater Bangkok and, additionally, the nature of work outside Bangkok was more static (such as work in well-established industries).

Provincial economic development did not guarantee the extension of SSS coverage amongst formal sector migrants. Therefore, mechanisms that ensure timely (and real-time) enrolment in the insurance scheme is needed. The analysis on SSS coverage amongst all migrants against provincial economy demonstrated a positive relationship as presented by spatiotemporal analysis. This information might be instructive for policy makers or public health practitioners to pay more attention to the economically less well-off provinces or some regions that displayed low SSS coverage.

Although not the prime objective of this study, we also found that not all migrant workers were insured, including those in the formal sector where SSS is mandatory. The lack of insurance coverage means that migrants are at risk of catastrophic payment for healthcare and health impoverishment. The risk of facing healthcare catastrophe and impoverishment is likely to be aggravated if migrants earn low wages, and they are staying in areas with high living expense (such as Greater Bangkok and the central region). Suphanchaimat et al., reported that SSS and the HICS helped reduce out-of-pocket (OOP) expenditure from healthcare utilisation amongst migrant workers in Southern Thailand [22]. However, the mentioned research focused on a certain province. Hence, a nationwide study on the impact of insurance on financial wellbeing at the individual level is likely to be useful. In addition to this, future research should investigate the relationship between different economies and OOP at regional and provincial levels. 

Methodology-wise, this study has both strengths and limitations. For strengths, it is likely that this study extends the value of existing knowledge on the association between provincial economy and migrant insurance enrolment in Thailand by using spatiotemporal analysis. Many recent studies in the migrant health field in Thailand focused on individual access to healthcare [22,23,24] and, most of the time, prior research relied on traditional regression analysis while ignoring spatiotemporal effects. The results from this study as shown in Figure 5 and Figure 6 and Table 5 confirm that spatial data analysis offered fitter outcomes relative to the analysis without accounting for spatiotemporal effects. The application of spatiotemporal analysis or any other novel statistical tools on migrant data, apart from the insurance enrolment, (such as data on healthcare access or health outcomes) is encouraged to extend academic richness in the migrant health research arena.

However, some limitations remain. Firstly, since data sources that we utilised were originated from different sources, these data sources have not been synchronised. Thus, we preferred to apply the term “ratio” to reflect the percentage of SSS insurees to total migrants over the term “proportion” as we could not assure that the nominator was part of the denominator. To some extent, this point also reflects room for improvement of the data collecting system on migrants in Thailand, especially for informal sector migrants who are still missing from public databases. Secondly, recorded data were not of sufficient duration to capture long-term change in the time trend. The completed economic data started from 2015 to 2018 but after 2018, the data (as available from the public domain) had not been updated. Regular monitoring of SSS coverage in parallel with economic information should be conducted. Despite a short-observed period, the readers may also benefit from this study as it serves as an example for how to explore insurance coverage and macro-economic indicators by a spatiotemporal approach. A short time frame means that the power of analysis is undermined, but as we recruited all provinces in the analysis, the issue of generalisability may not be a serious concern. Lastly, there were some unobserved variables which could affect the results of this study such as the difference of intrinsic behaviour or culture of migrants across provinces and even in-province policy changes (that might be reported or captured by the central government). These variables might not be captured by the routine quantitative reporting system. A qualitative study to draw lessons from remarkably high or low migrant insurance coverage is likely to help complement the quantitative results displayed in this study.

## 5. Conclusions

The ratio of SSS insurees to the formal sector changed overtime with greater density in western and southern regions, while the ratio of SSS migrants to all migrants was relatively high in the central and western regions including some provinces in the southern region. These ratios were extended in areas outside Greater Bangkok. Economic prosperity at provincial levels displayed different effects between these two indicators. A log-Baht increment in GPP per capita did not have significant association with SSS coverage amongst migrants in the formal sector, but a positive relationship was found in the ratio of insured migrants to all migrants. The contribution of some industries such as construction and agriculture, forestry, and fisheries exhibited a negative relationship with the interested indicators, but most industries did not demonstrate statistical significances. Methodology-wise, SDM presented the best fit compared with NB regression. Qualitative studies taking into account variables such as the intrinsic behaviour and culture of migrants and the dynamics of in-province employment policies and research that investigates the relationship between provincial economy and OOP are recommended.

## Figures and Tables

**Figure 1 ijerph-19-00181-f001:**
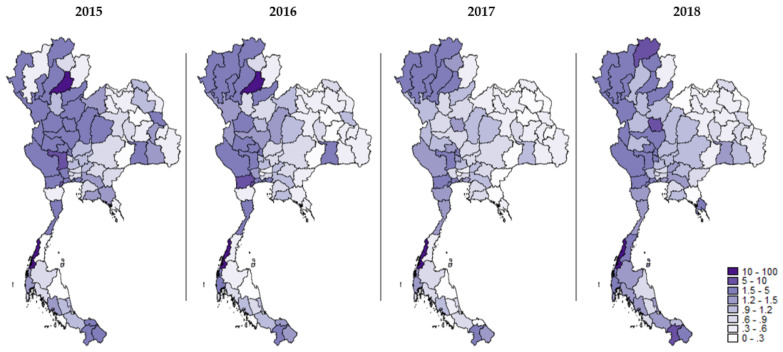
Geographical distribution of the ratio of insured migrants to formal-sector migrants in each year during 2015–2018.

**Figure 2 ijerph-19-00181-f002:**
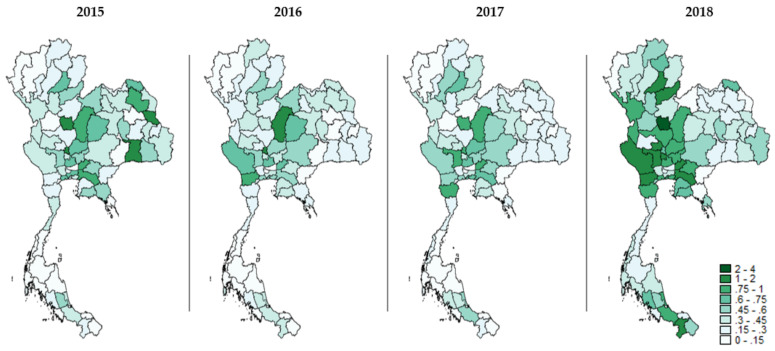
Geographical distribution of the ratio of insured migrants to all migrants in each year during 2015–2018.

**Figure 3 ijerph-19-00181-f003:**
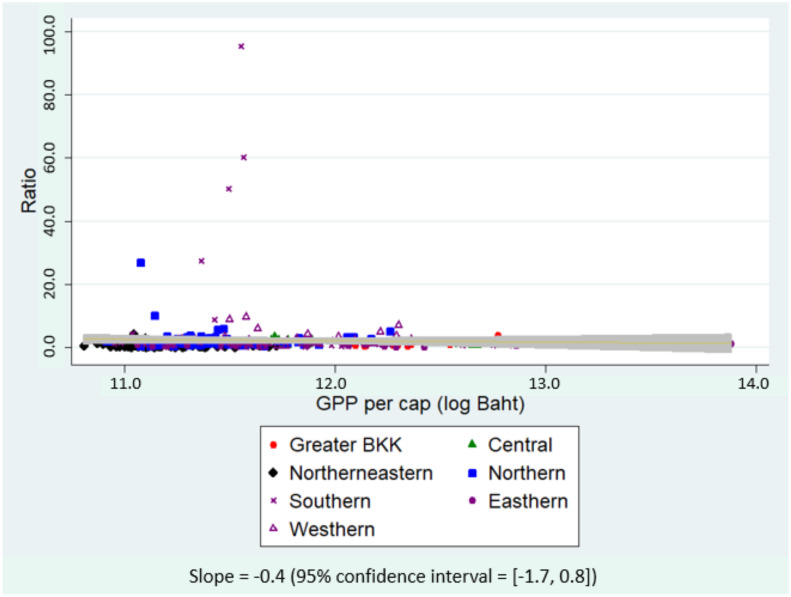
Scatter plot between the ratio of insured migrants to formal-sector migrants and GPP per capita, 2015–2018.

**Figure 4 ijerph-19-00181-f004:**
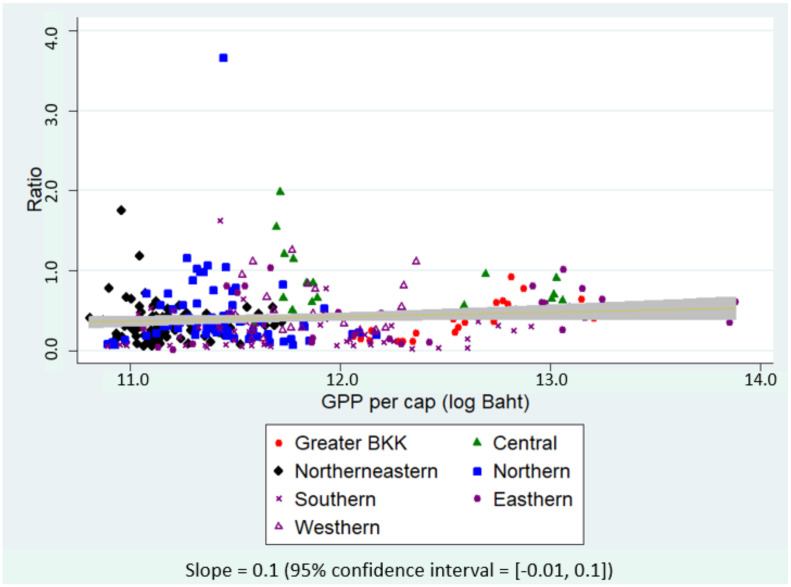
Scatter plot between the ratio of insured migrants to all migrants and GPP per capita, 2015–2018.

**Figure 5 ijerph-19-00181-f005:**
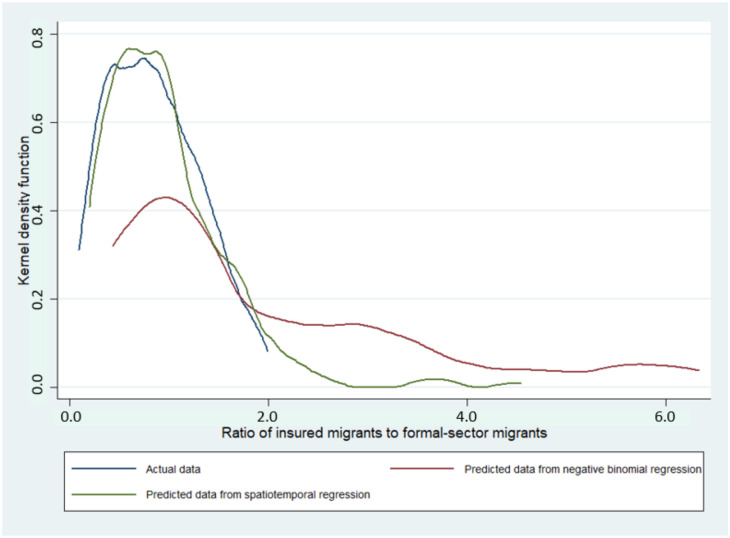
Kernel density plot of the ratio of insured migrants to formal-sector migrants from NB regression and spatiotemporal model and the actual data.

**Figure 6 ijerph-19-00181-f006:**
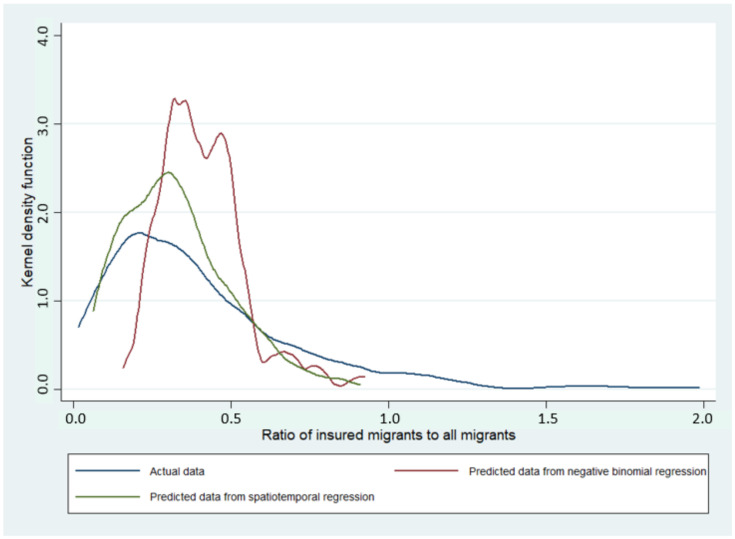
Kernel density plot of the ratio of insured migrants to all migrants from NB regression and spatiotemporal model and the actual data.

**Table 1 ijerph-19-00181-t001:** Number of migrants by regions, 2015–2018.

Year	2015	2016	2017	2018	Average Annual Growth Rate
Total migrants					
Greater BKK	656,725	763,88	855,750	1,248,091	30%
Central	36,612	39,868	40,218	49,637	12%
Northeastern	23,029	22,332	26,851	50,538	40%
Northern	174,100	174,065	152,936	171,226	−1%
Southern	312,787	293,428	292,044	361,694	5%
Eastern	211,343	154,124	164,835	260,373	8%
Western	91,938	75,195	74,298	128,528	13%
Formal-sector migrants					
Greater BKK	209,940	273,150	345,485	510,671	48%
Central	19,938	23,346	26,971	39,866	33%
Northeastern	12,837	12,803	18,472	33,421	53%
Northern	32,987	18,477	24,021	29,757	−3%
Southern	56,547	59,984	65,680	74,719	11%
Eastern	76,486	75,293	96,612	153,571	34%
Western	13,377	19,240	21,795	44,129	77%
Formal-sector migrants with the SSS					
Greater BKK	279,357	277,770	291,076	622,012	41%
Central	22,524	21,044	23,398	49,283	40%
Northeastern	9483	8458	9503	20,619	39%
Northern	34,122	32,747	33,508	73,478	38%
Southern	45,356	45,759	50,113	127,896	61%
Eastern	73,311	66,800	67,33	159,965	39%
Western	27,963	35,428	34,804	85,256	68%

**Table 2 ijerph-19-00181-t002:** Descriptive statistics of the dependent variables and the predictor variables at provincial level, 2015–2018.

Year	2015	2016	2017	2018
Insured migrants to formal-sector migrants				
Mean (sd)	3.6 (9.9)	2.5 (7.2)	2.0 (7.3)	2.9 (11.4)
Median (iqr)	1.2 (2.2)	0.9 (1.4)	0.8 (0.9)	1.1 (1.1)
Min/Max	0.0/114.6	0.0/78.7	0.0/80.8	0.1/133.6
Insured migrants to all migrants				
Mean (sd)	0.8 (2.7)	0.5 (1.2)	0.5 (1.0)	0.6 (0.5)
Median (iqr)	0.3 (0.4)	0.3 (0.3)	0.3 (0.4)	0.4 (0.5)
Min/Max	0.0/51.8	0.0/24.2	0.0/16.1	0.0/4.5
GPP per capita (THB)				
Mean (sd)	146,575.7 (141,654.5)	154,341.0 (146,360.6)	163,044.7 (155,287.0)	169,506.6 (162,109.2)
Median (iqr)	94,772.0 (93,987.0)	98,141.0 (102,399.0)	105,345.0 (112,669.0)	107,505.0 (104,095.0)
Min/Max	49,288.0/959,678.0	54,957.0/972,955.0	55,661.0/1,038,355.0	58,370.0/1,067,449.0
Agriculture, forestry and fishing (% GPP per capita)				
Mean (sd)	15.6 (10.6)	14.9 (10.0)	14.7 (9.8)	14.8 (9.8)
Median (iqr)	13.3 (14.7)	12.9 (15.0)	12.7 (15.0)	12.9 (14.3)
Min/Max	0.5/42.4	0.5/38.2	0.4/40.6	0.4/39.5
Manufacturing (% GPP per capita)				
Mean (sd)	18.8 (24.4)	19.1 (24.1)	19.4 (24.3)	19.1 (24.5)
Median (iqr)	8.4 (15.2)	9.2 (13.9)	8.9 (14.3)	8.5 (15.4)
Min/Max	0.9/112.0	0.8/110.6	0.7/104.1	0.6/103.0
Construction (% GPP per capita)				
Mean (sd)	3.2 (2.9)	3.2 (2.9)	3.1 (2.9)	3.1 (2.8)
Median (iqr)	2.4 (2.4)	2.4 (2.4)	2.2 (2.2)	2.2 (2.1)
Min/Max	0.4/19.6	0.4/18.8	0.3/19.2	0.4/18.5
Wholesale and retail trade and repair of motor vehicles (% GPP per capita)				
Mean (sd)	11.5 (20.5)	12.1 (21.6)	12.6 (22.1)	13.0 (23.0)
Median (iqr)	7.3 (7.1)	7.7 (8.2)	8.1 (8.2)	8.5 (8.7)
Min/Max	1.7/179.9	1.8/189.4	2.0/193.3	1.8/200.7
Accommodation and food service activities (% GPP per capita)				
Mean (sd)	2.7 (8.3)	2.8 (8.8)	3.1 (9.8)	3.4 (10.4)
Median (iqr)	0.4 (1.2)	0.4 (1.4)	0.5 (1.4)	0.5 (1.6)
Min/Max	0.0/65.6	0.0/69.2	0.0/78.4	0.0/83.0

Note: sd = standard deviation; iqr = interquartile range.

**Table 3 ijerph-19-00181-t003:** Association between dependent variables (ratio of insured migrants to formal-sector migrants, and ratio of insured migrants to all migrants) and all predictor variables by negative binomial regression.

Dependent Variables	Ratio of Insured Migrants to Formal-Sector Migrants			Ratio of Insured Migrants to All Migrants		
	IRR	95% CI	*p*-Value	IRR	95% CI	*p*-Value
GPP per capita—log Baht	0.817	0.569–1.171	0.271	0.915	0.685–1.223	0.550
Region (reference = Greater Bangkok and its vicinity)						
Central	1.154	0.617–2.157	0.653	2.667	1.644–4.328	<0.001
Northeastern	0.583	0.304–1.118	0.104	1.391	0.830–2.332	0.210
Northern	2.226	1.162–4.265	0.016	1.872	1.118–3.135	0.017
Southern	4.888	2.540–9.405	<0.001	1.119	0.658–1.901	0.679
Eastern	0.874	0.490–1.560	0.649	1.651	1.046–2.606	0.031
Western	3.036	1.561–5.906	0.001	2.189	1.291–3.713	0.004
Agriculture, forestry, and fishing—% GPP per capita	0.983	0.967–0.999	0.042	0.998	0.985–1.011	0.773
Manufacturing—% GPP per capita	0.999	0.989–1.009	0.823	1.007	0.999–1.015	0.088
Construction—% GPP per capita	0.963	0.884–1.049	0.385	0.953	0.891–1.020	0.165
Wholesale and retail trade and repair of motor vehicles—% GPP per capita	1.025	1.010–1.041	0.001	1.011	0.999–1.023	0.081
Accommodation and food service activities—% GPP per capita	0.946	0.917–0.976	<0.001	0.985	0.963–1.008	0.209

Note: IRR = incidence rate ratio; CI = confidence interval.

**Table 4 ijerph-19-00181-t004:** Association between dependent variables (ratio of insured migrants to formal-sector migrants and ratio of insured migrants to all migrants) and all predictor variables by spatiotemporal regression.

Dependent Variables	Insured Migrants in Formal Sector			Insured Migrants in All Sectors		
	IRR	IRR 95% CI	*p*-Value	IRR	IRR 95% CI	*p*-Value
GPP per capita—log Baht	0.637	0.301–1.352	0.240	1.682	1.065–2.656	0.026
Region (reference = Greater Bangkok and its vicinity)						
Central	1.703	0.533–5.436	0.369	3.931	1.616–9.566	0.003
Northeastern	1.742	0.524–5.788	0.365	7.015	1.823–26.985	0.005
Northern	2.160	0.637–7.327	0.217	4.601	1.448–14.624	0.010
Southern	0.222	0.015–3.383	0.279	6.100	1.448–25.691	0.014
Eastern	2.143	0.550–8.353	0.272	11.313	2.692–47.538	0.001
Western	2.937	1.041–8.284	0.042	5.598	2.448–12.799	<0.001
Agriculture, forestry, and fishing—% GPP per capita	0.999	0.975–1.022	0.906	0.995	0.977–1.012	0.543
Manufacturing—% GPP per capita	1.009	0.995–1.022	0.216	1.002	0.993–1.010	0.713
Construction—% GPP per capita	0.957	0.866–1.057	0.388	0.991	0.896–1.097	0.867
Wholesale and retail trade and repair of motor vehicles—% GPP per capita	0.997	0.975–1.018	0.754	1.015	0.997–1.033	0.107
Accommodation and food service activities—% GPP per capita	1.007	0.959–1.057	0.788	0.981	0.946–1.018	0.316

Note: IRR = incidence rate ratio; CI = confidence interval.

**Table 5 ijerph-19-00181-t005:** Assessment of the model fitness for NB regression and the spatiotemporal regression.

Assessment	Insured Migrants to Formal-Sector Migrants		Insured Migrants in All Migrants	
	NB Regression	Spatiotemporal Regression	NB Regression	Spatiotemporal Regression
MAE	2.0	0.8	0.2	0.1
MSE	48.3	14.4	0.1	0.1
RMSE	7.0	3.8	0.3	0.3
MAPE	189.9	34.8	97.7	36.9
AIC	5336.8	628.0	5146.8	609.5

Note: MAE = mean absolute error; MSE = mean squared error; RMSE = root mean squared error; MAPE = mean absolute percent error; AIC = Akaike Information Criterion.

## Data Availability

All data were retrieved from the available public domain as mentioned in Section 2.

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
