# Peer review of "Spatial Panel Data Analysis on the Relationship between Provincial Economic Status and Enrolment in the Social Security Scheme amongst Migrant Workers in Thailand, 2015–2018"

_ijerph, 2021, doi:10.3390/ijerph19010181_

Round 1

Reviewer 1 Report

Thank you for giving me the opportunity to review this paper. The paper addresses an interesting topic but I think that the work needs a major revision in some aspects, especially the introduction, methodology and discussion.

I think that one of the weakness of the study has been the scant description of the methodology. Unfortunately, it does not allow the reader to understand in detail some of the results obtained. Although the results are interesting they need to be supported by a clear methodological description.

In my opinion, it would be advisable for the authors to improve the introduction and allow the reader to know some details about the phenomenon of migration in Thailand. It would be very interesting to know what the migrant population is like (a brief description in demographic and social terms). Issues like who can access the Social Security Scheme, who can apply for these benefits, how long they have to wait. How lack of access healthcare affects their health. Is access to health care public and universal?

Another section that need a revision is the discussion, it would be interesting to know what implications not having a Social Security Scheme has for immigrants, or health situation in relation to salary. Also, what are the differences between the different economic sectors in terms of health and the impact of this differences in terms of out-of-pocket expenses if not covered. Finally, what regions are the most affected.

Some minor comments:

  1. Introduction

On page 1, line 33, the authors say “Migrant health is therefore recognized as a global public health priorities [sic]”. In my opinion, it is an important sentence that, nevertheless, lacks argumentation and connection with the rest of the text. It would be recommended that the authors try to better connect and argue this aspect with the rest of the article.

Between lines 34 to 38, it would be appreciated if the authors indicate the percentage of the immigrant population with respect to the total Thai population, and also if they describe the percentage of the immigrant population from each country (Cambodia, Lao PDR, Myanmar, etc. ) so the reader can have an idea of ​​the population weight with respect to the total population of Thailand.

  1. Materials and Methods

2.1 Study design

I would suggest the authors to better explain this section:

“We used quantitative secondary data analysis on time-series data during 2015-2018. The unit of analysis was a panel of 77 provinces in Thailand. This meant we analyzed a total of 308 observations. Data in 2019 and onward were not complete enough to perform 56 the analysis.”

 In it, it is not very clear what the “panel of 77 provinces in Thailand” is, or why they analyze 308 observations, or where do these observations come from.

I would also suggest the authors to use the expression “two indicators” instead of “main outcomes” and describe in a formula the numerator and denominator of the indicator obtained (page 2, line 63).

The sentence “Therefore, it was possible that the numerator was not part of the denominator” (page 2, line 67), does not seem necessary.

On page2, line 67 to 79, it is necessary to describe in detail what the dependent variable and independent variables will be. In particular, it is necessary to know the operational definition of the independent variables that are not described in detail. Eg “regional domicile”, how is it measured? Is it a percentage of migrant persons residing in those regions? Similarly, the rest of variables are in need of a more detailed description in the text or in a supplementary section since it is important to know the detailed information of each variable so that the results can be comparable with other studies.

2.3 Data analysis

It would be advisable for the authors to briefly explain what effects the following sentence refers to: "there was a possibility that one province might have a spill over effect to nearby provinces." (page 3, lines 93 to 94).

The Spatial Durbin Model appears correct, but cannot be stated with certainty since bibliographic reference 12 is mis-referenced or has incomplete information.

  1. Elhorst JP. Spatial panel models Netherlands University of Groningen, Department of Economics, Econometrics and Finance 2011.

In general terms, it would be advisable to improve the description of the methodology and indicate what software has been used for data analysis.

  1. Results

It would also be advisable to indicate how the “Average annual growth rate” has been calculated and indicate it in Table 1. With respect to which year has the annual growth rate been calculated? At the moment I do not have enough information to assess whether the results presented in Table 1 and described between rows 116 and 123 (page 3) are correct, when describing the growth, it would be advisable to say with respect to which year this population growth has been observed.

Finally, the title of Figures 1 and 2 seem to refer to a period, from 2015 to 2018. However, the map do not depict a whole period but year by year.  Therefore, the title should state all years (2015, 2016, 2017 and 2018)

Author Response

Dear reviewer,

Please find the responses to all comments in the attached file.

Best regards,

Shaheda

Reviewer 2 Report

There are only 4 observations for each time series (years 2015-2018). The series are therefore too short for any analysis based on regression models. Only spatial analysis can be performed on such data. No spatio-temporal or temporal analyses can be performed. The models obtained are therefore unreliable, as they cover only 4 years. No conclusions can be drawn from them. The method itself is good, but not for such data. The time series should be extended backwards or forwards, or other models should be used. Only spatial analysis in subsequent years can be carried out on the basis of these data.

Author Response

Thank you so much, we are also concerned about this point. However, the completed data was available in only four years as we mentioned in the limitations. Therefore, the unit of analysis was a panel of 77 provinces in four years (2015–2018) to get 308 records for the study (77 provinces x 4 years).

We also add some sentences related to this issue in the limitation as followed (page 13, lines 409-413).

Despite a short-observed period, the readers may also benefit from this study as it serves as an example how to explore the insurance coverage and macro-economic indicators by spatiotemporal approach. A short time frame means the power of analysis is undermined but as we recruited all provinces in the analysis, the issue of generalisability may not be a serious concern.  

Round 2

Reviewer 2 Report

The Authors partially revised their manuscript. Although it still has some flaws, they tried to explain the doubts. Therefore now the manuscript can be published in its present form.